# Morphological and Molecular Characterization of Three Endolichenic Isolates of *Xylaria* (Xylariaceae), from *Cladonia curta* Ahti & Marcelli (Cladoniaceae)

**DOI:** 10.3390/plants8100399

**Published:** 2019-10-08

**Authors:** Ehidy Rocio Peña Cañón, Margeli Pereira de Albuquerque, Rodrigo Paidano Alves, Antonio Batista Pereira, Filipe de Carvalho Victoria

**Affiliations:** 1Grupo de Investigación Biología para la Conservación, Departamento de Biología, Universidad Pedagógica y Tecnológica de Colombia, Avenida Central del Norte 39-115, 150003 Tunja, Colombia; ehidy.pena@uptc.edu.co; 2Núcleo de Estudos da Vegetação Antártica (NEVA), Universidade Federal do Pampa (UNIPAMPA), Avenida Antônio Trilha, 1847, 97300-000 São Gabriel CEP, Brazil; neva@unipampa.edu.br (M.P.d.A.); antoniopereira@unipampa.edu.br (A.B.P.); 3Max Planck Institute for Chemistry, Andre Araujo Avenue, 2936, 69067-375 Manaus, Brazil; rodrigo.alves@mpic.de

**Keywords:** fungi, phylogeny, lichen, ITS, qPCR, Brazil, *Xylaria berteri*, *Xylaroideae*

## Abstract

Endophyte biology is a branch of science that contributes to the understanding of the diversity and ecology of microorganisms that live inside plants, fungi, and lichen. Considering that the diversity of endolichenic fungi is little explored, and its phylogenetic relationship with other lifestyles (endophytism and saprotrophism) is still to be explored in detail, this paper presents data on axenic cultures and phylogenetic relationships of three endolichenic fungi, isolated in laboratory. *Cladonia curta* Ahti & Marcelli, a species of lichen described in Brazil, is distributed at three sites in the Southeast of the country, in mesophilous forests and the Cerrado. Initial hyphal growth of *Xylaria* spp. on *C. curta* podetia started four days after inoculation and continued for the next 13 days until the hyphae completely covered the podetia. Stromata formation and differentiation was observed, occurring approximately after one year of isolation and consecutive subculture of lineages. Phylogenetic analyses indicate lineages of endolichenic fungi in the genus *Xylaria*, even as the morphological characteristics of the colonies and anamorphous stromata confirm this classification. Our preliminary results provide evidence that these endolichenic fungi are closely related to endophytic fungi, suggesting that the associations are not purely incidental. Further studies, especially phylogenetic analyses using robust multi-locus datasets, are needed to accept or reject the hypothesis that endolichenic fungi isolated from *Xylaria* spp. and *X. berteri* are conspecific.

## 1. Introduction

Endophyte biology is a recent science that explores the diversity and ecology of microorganisms that live in several kind of hosts, as plants, fungi, and lichens, those belonging to the latter are known as endolichenics, a group of endosymbionts that reside within symptomless lichen thalli, in preferential association with green algal photobionts [1,2,3,4,5]. They inhabit the intercellular spaces of their hosts [6], their colonization is asymptomatic, they are hyperdiverse, and many are transmitted horizontally analogous to plant endophytes [1,5,7,8]. In the last thirty years, endolichenic fungi have been known to occur in every lichen species sampled in ecosystems ranging from hot deserts to moist forests and the arctic tundra [1,2,3,8,9,10,11,12,13].

Efforts have been made to compare the different fungi that inhabit the tissues of lichens and their hosts [2,5], thus establishing effective methods for surface sterilization in order to estimate their diversity [3,10,12]. Similarly, some studies have attempted to explain both the evolution of endophytism and the diversification of Ascomycota endolichenic species [1], and also to evaluate the biotic, biogeographic, and abiotic factors that structure their communities [8,13]. Likewise, research has been carried out on the metabolites produced [6] by endolichenic fungi and to evaluate their antimicrobial potential [11]. Recently, a review of molecular phylogenetic relationships among endophytic and endolichenic fungi of the family Xylariaceae was presented, showing that the great majority of isolates were included within the Hypoxyloideae and Xylarioideae subfamilies [9]. In addition, knowledge about the role of non-mycobionte endolichenic fungi in the lichenization process is still scarce [14,15], unlike that established for bacteria associated with lichen thalli, which, in some cases, are already known to have an active role in the lichen assembly [14,16]. Thus endolichenic fungi could also participate in this process, taking the lichen concept beyond the classic concept of specificity and selectivity of the mycobiont and photobiont compartments [16]. This can be assessed from re-synthesis and gene expression experiments, as there are protocols aimed at re-cultivating previously isolated mycobionts and photobionts in a new in vitro lichen [16], and recent studies suggest the participation of specific genes in this initial fungal and algal recognition process for eventual lichenization, such as the *HPPD* gene (the algae putative 4-hydroxyphenylpyruvate dioxygenase) and the D-arabitol dehydrogenase gene in fungi [17].

To date, studies on endolichenic fungi in India from foliose and fruticose lichen species belonging to five families (Caliciaceae, Collemataceae, Lobariaceae, Parmeliaceae, Physiaceae, Ramalinaceae, and Roccellaceae) have reported 24 species [2] and 33 morphospecies [3] from its tropical regions. In contrast, Tripathi et al. [4,12] worked on the endolichenic fungi of the temperate regions of Kumaun, Himalaya and isolated 10 species that are reported as true endolichenic fungi. In China, endophytic fungi from seven lichen species belonging to five families (Cladoniaceae, Parmeliaceae, Ramalinaceae, Teloschistaceae, and Verrucariaceae) were investigated, with a total of 32 taxa [10]. In Sri Lanka, 29 endolichenic fungal strains were isolated from the lichens *Parmotrema* sp., *Usnea* sp., and *Pseudocyphellaria* sp. However, this number is small when compared with those of some more studies from North American states [1,5,8,9,18]. A bibliographic review showed that the present work is the first study on the diversity of endolichenic fungi harboured in lichen thalli from South America. 

The genus *Xylaria* Hill ex Schrank is one of the most diverse within the family Xylariaceae, with about 600 species that are mostly tropical, and has been shown to be paraphyletic [19,20]— along with 85 other genera of the family, it is classified within the class Sordariomycetes [21]. Members of this genus are considered important saprophytes, sometimes from slight to strongly parasitic species found in the wood of trees and even on leaves and rarely on fruits [19,22,23]. *Xylaria* are visible during sexual sporulation, forming relatively large, macroscopic stromata [20]. Despite their saprophytic condition, *Xylaria* species are an example of typical endophytic fungi isolated from the fragments of apparently healthy plants and lichens [1,9,24,25,26] and are predominant in studies on the diversity of both endolichenic and endophytic fungi [3,27,28]. 

Xylariaceae often remain sterile in culture or reproduce only asexually [21]; therefore, in the absence of teleomorphic features, cultures can in some cases be classified based on the colonial and anamorphic features observed, such as growth rate, colour, colony surface morphology, as well as conidiophore branching and the nature of conidiogenous cell proliferation [21,29,30,31,32]. Moreover, molecular techniques have been employed in the detection and identification of endolichenic fungi, principally using Internal transcribed spacer (ITS) regions and the β-tubulin gene to analyse the phylogenetic positions or relationships at species or interspecies levels [1,9,19,22].

*Cladonia curta* Ahti & Marcelli is a squamulose lichen that grows on decomposing logs at the edge of the forest and is considered a very rare or ignored species. In Brazil, this species is distributed at four sites in the southeast and southern of the country, in mesophilous forests and the Cerrado in the states of Minas Gerais, and São Paulo, in the Federal District of Brasília, between 800–1200 m above sea level [33] and in the Pampa biome of Rio Grande do Sul state [33]. Considering the little explored diversity of endolichenic fungi, there is still much to be understood about their features and the particularity and conditions of their association with symbionts within lichen thalli; an immense amount of effort and additional studies are still required to increase their knowledge and to understand their phylogenetic relationship with other endophytic fungi isolated from plants and lichens. Thus, this study aimed to molecularly and morphologically identify three endolichenic fungi isolated from *C. curta*, collected from the south of Brazil and to determine their phylogenetic relationship with other endophytic fungi.

## 2. Results

### 2.1. Isolation of Endolichenic Fungi

We characterized three endolichenic fungi isolated from 30 thalli fragments of *Cladonia curta* sampled in São Gabriel, Brazil (voucher HBEI 023), using both molecular and morphological features. The initial hyphal growth of the isolated *Xylaria* spp. on *C. curta* podetia started four days after inoculation and continued for the next 13 days until the hyphae completely covered the podetia (Figure 1A,B). The podetia were transferred to petri dishes with fresh culture medium to continue the growth and develop mycelium until colony formation (Figure 1C). The appearance of colonies of *Xylaria* spp. isolates on the MS medium in the photoperiod and in darkness is shown in the Appendix A (Appendix A). Stromata formation and differentiation occurred after ten months (isolates MS1 and LB1) and one year (isolated MS2) of isolation and consecutive subculture of lineages.

### 2.2. Phylogenetic Analyses 

The sequences obtained from axenic cultures of endolichenic fungi from *Cladonia curta* resulted in BLASTn hits close to xylariaceous fungi. The isolates were considered as belonging to the genus *Xylaria* after comparison of their nucleotide sequences revealed an identity greater than 98% and 97% for the ITS regions of rDNA and β-tubulin, respectively. The compatibility of the endolichenic fungal species was mainly with the sequences deposited as *Xylaria berteri* and reported as endophytic fungi from lichens [34], angiosperms [19,27,29,35,36,37,38,39,40,41,42], liverworts [24], and ferns [42] (Table 1). 

The total number of 52 sequences of ITS regions of rDNA (Table 1.) were compared with the sequences from our three endolichenic fungi lineages, obtained after isolation procedures (for more details see the methods sections). The maximum likelihood (Figure 2) and Bayesian inference (Figure 3) trees based in this dataset were congruent and differed slightly in their topology. Based on our results, four distinct clades are supported (A, B, C, and D). Clade A comprises a total of 14 endolichenic fungi sequences from species including *Xylaria arbuscula* Sacc., *Xylaria venustula* Sacc., seven species of *Xylaria* isolated from six species of lichens (*Cladonia evansii*, *Usnea subscabrosa*, *Cladonia subradiata*, *Flavoparmelia caperata*, and *Cladonia didyma*) [9], and four species of endolichenic fungi isolated from *Peltigera neopolydactyla* s.l. lichen described by Arnold et al. [1]. The formation of this clade is supported by both the maximum likelihood and Bayesian analyses with bootstrap values of 1% and a posterior probability of 0.25. An endophytic fungus of *Faramea occidentalis* (*Rubiaceae*) (KT289626) was included in this clade using BI and was inferred as a sister group of two species sequences reported as endolichenic fungi (PP = 0.21). 

The comparison of phylogenetic trees obtained from the maximum likelihood analyses and Bayesian Inference, showed that clade B formation presents slight differences in the composition and grouping of species. This clade included the new sequences from this study and the sequences of fungi principally reported as endophytic and some endolichenic. Of these taxa, nine were identified as *Xylaria berteri* in GenBank and were isolated from angiosperms in different studies: six sequences were obtained from fungi isolated from soybean cultivars *Glycine max* (*Fabaceae*) by De Souza Leite et al. [38], two from dicotyledons by Hsieh et al. [19], and one from *Myrceugenia ovata* var. nanophylla (Myrtaceae) by Vaz-Aline et al. [29]. The remaining species that comprise clade B include three species of endophytic fungi from liverworts that correspond to two species of *Xylaria* isolated from *Bazzania* sp. (*Lepidoziaceae*), *Trichocolea tomentella* (Ehrhart) Dumortier (*Trichocoleaceae*), and an unidentified isolate from *Plagiochila* (*Plagiochilaceae*) (EU686041), as well as one sequence of endophytic *Xylaria* in *Cinchona pubescens* (*Rubiaceae*), two endophytic fungi of the *Xylariaceae* family grown from the interior of healthy leaves of *Coffea arabica* (*Rubiaceae*), two isolated endophytic fungi from *F. occidentalis* and *Arundinaria gigantea* (*Poaceae*), respectively and three endolichenic fungi isolated from the lichens *Lecanora oreinoides* (JQ761659), *Usnea mutabilis* (JQ760593), and *Umbilicaria mammulata* (KT289568). This clade showed low ML support (1%) and was moderately supported on BI (PP = 0.49). Bayesian analysis resulted in a modification of the clade B shown by ML, excluding *F. occidentalis* and an endolichenic species from *P. neopolydactyla* s.l. (KT289550), which were grouped with the hepatic endophytic species that comprise clade C.

The isolate *Xylaria* sp.1 (LB1) was grouped together with an endophytic fungus identified as *Xylaria berteri*, cultivated from soybean leaves of the “Conquista” cultivar in a clade poorly supported by the maximum likelihood method (7%). Similarly, the posterior probability that *Xylaria* sp.1 is closest to the soybean endophytic species was low (0.08). The culture *Xylaria* sp.2 (MS1) was shown to form a clade with one endophytic fungus of *F. occidentalis* (KT289626) and an endolichenic fungus cultured from *P. neopolydactyla* s.l. (KT289550) (ML = 3%). Using BI, this isolate showed a close relationship with seven sequences of endophytic fungi from angiosperms identified as *X. berteri.* The isolate *Xylaria* sp.3 (MS2) was inferred as a sister of the other lineages (PP = 0.38). Sequence similarities by ML between the isolate *Xylaria* sp.3 (MS2) and the taxon identified as *X. berteri* (GU324750) isolated from the bark of a dicotyledonous tree was 6%. Similarly, this isolate was grouped within the same clade as that of the endophytic *X. berteri* from *M. ovata* var. *nanophylla*, supported with a posterior probability value of 0.07. Lower sequence similarities of ML analysis were found between *Xylaria* sp.3 (MS2) and the taxon identified as *X. berteri* (GU324750) isolated from the bark of a dicotyledonous tree, up to 6% from both homologues. Likewise, this isolate was grouped within the same clade as that of the endophytic *X. berteri* of *M. ovata* var. *nanophylla*, supported with a posterior probability value of 0.07.

Seven species of *Xylaria* isolated by Davis et al. [43] from liverwort rhizoids of *Bazzania* species and three species of endolichenic fungi from *P. neopolydactyla* s.l. comprise clade C. This clade has low support with a bootstrap value of 28% and a posterior probability of 0.18. The clade D comprises four endolichenic fungi isolated from *P. neopolydactyla* s.l. and a *Xylaria* isolate from *Bazzania* sp. with weak bootstrap support (18%) but strong posterior probability support (0.99); however, the sequence of *Xylaria* sp. (AY315388) was not included in this clade.

In total, 43 sequences of the β-tubulin gene (Table 1) were compared to the sequences obtained from both of our lineages (LB1 and MS1, for more details see the methods sections). Two sequences were mining from the BLASTn search, one sequence was selected from *Xylaria berteri* cultivated by Hsieh et al. [19], and the remaining sequences correspond to species of endolichenic fungi isolated and cultivated by U’Ren et al. [9]. All sequences are available on the public databases and its accession number are indicated in the Table 1. The fungal isolate MS2 was not included in the phylogenetic analysis because its sequence presented a smaller size, low quality, and an e-value > 0 in the BLASTn query that generated a conflict during alignment with the other sequences of endophytic and endolichenic fungi. Considering the results generated by the phylogeny of ITS regions of rDNA, we considered it pertinent to include the sequence of *X*. *berteri* (GQ502698) grown in a culture from ascospores. The trees generated by the ML and BI analyses based on the β-tubulin dataset were highly similar in topology. Since these trees were congruent, only the tree obtained by Bayesian inference (Figure 4) is presented and described in this document. 

The phylogenetic tree obtained, clearly showed the formation of two large clades and the sequence of an endolichenic fungus from *Pseudevernia consocians* outside the groups. Of them, a clade with high support (PP = 0.83) included the species of endolichenic fungi isolated from *Cladonia curta* (LB1 and MS1), three sequences of *X. berteri*, and sequences of endophytic fungi mainly isolated from lichen species of the genus *Cladonia* (*C. evansii*, *C. didyma*, *C. subradiata*, *and C. subtenius*) and other lichens (*Usnea subscabrosa* and *Flavoparmelia caperata*). The remaining species of endolichenic fungi formed a branch with low posterior probability support (0.35). The sequences of the fungal isolates (MS1 and LB1) were closely related to three sequences of *X. berteri*, two saprophytic and one endophytic fungus (PP = 1), sequences of *Xylaria cubensis* (Mont.) Fr. and *Xylaria* cf. *heliscus*, and endolichenic fungi from *Lecanora oreinoides* and *Usnea mutabilis* (PP = 1). This clade was strongly supported with a posterior probability value of 1. 

### 2.3. Phenotypic Characterization of Isolates in Culture Medium

Isolate: *Xylaria* sp. 1 (LB1—Figure 5)

Culture: colonies on MS medium cover a 9 cm plate in 18 days. In photoperiod of 16 h light and 8 h dark, at first white, cottony, zoned in the centre, and irregularly radial towards the edge; abundant aerial mycelium forming prominent protrusions with flattened topography and finally forming irregular beige dark spots (M&P K1–Plate 9). Reverse mostly remaining white for up to four weeks, finally turning yellowish white (M&P G1–Plate 11); radial with black scores up to one millimetre in diameter forming concentric zones. In the centre of the colonies was observed a primordium of stroma, pyramidal-acute, black with a cylindrical stipe up to 0.8 mm in height with irregular flattened branches. In the dark, the colonies were uniformly white, cottony, zoned in the centre, radial towards the margin with elevated topography. Reverse radial, uniformly yellowish white. Clear exudate droplets on the surface of young colonies and (M&P B9–Plate 13) droplets were also found in mature colonies. On 2% MEA the primordia grew and the mycelium formed more stromata.

Stromata: 10 solitary or gregarious stromata were formed; stipe initially cylindrical to clavate, later bifurcate to cerebriform with several small digitate projections, 1.5–3 mm (base) and 2–6 mm (apex) wide × 11–16 mm high, verruculose to dendroid surface; apex white and pale orange (M&P B9–Plate 4) at base during conidia formation to finally gray ochre (M&P E1–Plate 8) (Figure 5A–C). Hyphae with thick wall, dark brown, regularly septate with branches forming 90° angles, without reaction to the Melzer reagent (Figure 5E). The central axis of the stroma presents thin hyphae immersed in a gelatinous matrix. 

Conidiospore: ovoid to clavate, 1.72–2.96 (3.47) × 4.30–6.35 (8.22) μm (M = 5.66–2.41 μm) of thin wall, smooth ornamentation, hyaline with flattened basal scar (Figure 5G,H). Conidiophores laterally compressed into tight layer or palisade.

Perithecia: initial formation of primordial perithecium-like subglobose structures, not releasing pigments when in reaction with 5% KOH (Figure 5D).

Ascospores: only two ascospores found, both reniform, 2.27–2.20 × 5.01–5.10 μm, dark brown to blackish brown with thin wall showing smooth ornamentation and longitudinal straight germ slit (Figure 5F).

Specimens examined: BRAZIL, Rio Grande do Sul—São Gabriel. Endolichenic fungi isolated from *C. curta*. 11th February 2016. Isolator: Peña-Cañón, R. HBEI 001 and HBEI 021.

Commentary: based on the synoptic key to *Xylaria* species by Callan and Rogers [30] the isolate presents colonial and anamorphic features observed in cultures that are shared with *Xylaria multiplex* (Kunze) Fr. and *Xylaria longiana* Rehm.

Isolate: *Xylaria* sp. 2 (MS1—Figure 6) 

Culture: colonies on MS medium cover a 9 cm plate in 21 days. In a photoperiod of 16 h light and 8 h dark, the mycelium was uniformly white, cottony, radial, non-zoned with flat topography. Reverse radial and uniformly yellowish white (M&P G1–Plate 11). The colonies showed a primordium of stroma, not-central, pyramidal-obtuse, black and irregular in appearance and shape on the reverse. In the dark, the colonies were white with greyish white centre (M&P A1–Plate 39), cottony, radial, margin forming wide lobes, and irregular topography. Reverse yellowish white (M&P G1–Plate 11) becoming more pale-coloured towards the margin, except in an area developing partridge brown (M&P L12–Plate 15) to black patches. Clear exudate droplets exudate on the surface of young colonies, orange (M&P B12–Plate 13) droplets in mature colonies. Black immature stromata with cylindrical stipe up to 0.8 mm high, apex branched and irregularly flattened. On 2% MEA the primordial stromata grew and the mycelium formed more stromata.

Stromata: on MS medium, cylindrical stipe 12 mm high, central and irregularly branched in the apex; initially orange (M&P E8–Plate 13) later black at the base with white apex (Figure 6A). On 2% MEA, 17 irregular to irregularly clavate stromata, 1–4 mm (base) and 1–7 mm (apex) wide × 9–16 mm high were formed, verrucose to dendroid surface with distal lobes, irregularly verrucose, extended up to 5 mm from a base two to three millimetres high. Base of stipe white passing to willow brown colour (M&P L7–Plate 15) to smoke brown (M&P A2–Plate 16) and almost black with apex pale orange (M&P B9–Plate 4) (Figure 6B,C). Hypothecium of ectostroma formed by pseudo-catenulate, erect and varied bifurcated hyphae with thin wall, probably being the beginning of hymen formation; some hyphae presented anastomoses (Figure 6E).

Conidiospore: obovoid to clavate, 4.38–5.86 × 9.92–10.99 (13.35) μm (M = 5.21–10.60 μm, n = 20), with thin wall, smooth ornamentation, hyaline with flattened basal scar (Figure 6F–H).

Perithecia: initial formation of perithecium was observed but without ascospore development (Figure 6D). 

Specimens examined: BRAZIL, Rio Grande do Sul—São Gabriel. Endolichenic fungi isolated from *C. curta.* 11th February 2016. Isolator: Peña-Cañón, R. HBEI 002 and HBEI 022.

Commentary: based on a synoptic key to *Xylaria* species by Callan and Rogers [30], the isolate shares the greatest number of colonial and anamorphic features with *Xylaria longipes* Nitschke, *Xylaria polymorpha* (Pers.) Grev. and *Xylaria schweinitzii* Berk. & M.A. Curtis. 

Isolate: *Xylaria* sp. 3 (MS2—Figure 7) 

Culture: colonies on MS medium cover a 9 cm plate in 28 days. In a photoperiod of 16 h light and 8 h dark, at first white, cottony, non-zoned, margin forming wide lobes and elevated topography. The aerial mycelium later varied to pale pink (M&P D1–Plate 1) towards the margin. Reverse yellowish white with irregular zonation of semi-circular brown spots (M&P H5–Plate 7). The formation of stromata primordia was observed at prolonged incubation times. Towards the edge of the colony was observed some primordium of stroma with pyramidal-acute apex, varying from pale orange (M&P B9–Plate 4) to white and cylindrical stipe, orange (M&P E8–Plate 13), between 0.3–0.9 mm in height. In dark, colonies white, cottony, with finely lobed margins and flat topography. Reverse yellowish white (M&P G1–Plate 11) with black spots arranged radially.

Stromata: two stromata 0.6–0.9 mm in height with 1 mm diameter, smooth and orange (M&P A12–Plate 4) basal region and apex bifurcated, cottony, and whitish, covered with conidiophores (Figure 7A). 

Conidiospore: obovoid to clavate, 1.67–2.41 (3.65) × (4.10) 4.81–6.26 (6.66) μm (M = 2.19–5.45 μm, n = 20), with thin wall, smooth ornamentation, hyaline with flattened basal scar (Figure 7B). Conidiophores laterally compressed into a tight layer or palisade (Figure 7C).

Specimens examined: São Gabriel, Rio Grande do Sul, Brazil. Endolichenic fungi isolated from *C. curta*. 11 February 2016. Isolator: Peña-Cañón, R. HBEI 003.

Commentary: based on a synoptic key to *Xylaria* species by Callan and Rogers [30] the isolate shares the greatest number of colonial and anamorphic features with *Xylaria polymorpha* (Pers.) Grev. and *Xylaria longiana* Rehm. 

### 2.4. Re-Synthesis and qPCR Evaluation on the Endophytic Role in the Lichen Symbiosis

For the re-synthesis test of *Xylaria* sp. and the *Cladonia curta* photobiont, fungal disks were inoculated onto glass plates on MBB medium one-day prior to contact with the photobionte (6 December). The next day, the algae was spread 2.5 cm apart around the disk. The fungus growth started three days after the inoculation until the algae reach four days later. After contact, both the fungus and the algae continue their growth (Figure 8A–C). In the contact zones, the formation of a structure in which the color of the fungus and the alga varies (Figure 8C).

Regarding gene expression, it was observed that both genes tested showed, at least two-fold expression in the MS2 and LB1 strains. There were no significant differential expressions for the MS1 lineage. The Oxido gene showed increased expression at 4 days of the experiment, but this was not accompanied by the expression of the *HPPD* algae gene. For the LB1 lineage both genes tested were overexpressed on the first day of experiment, varying their expression over the 21 days. The *HPPD* gene showed significant increase of expression at 21 days for MS1 and at eight days for LB1 (Figure 9).

## 3. Discussion

Phylogenetic analyses (ML and BI) based on the ITS regions of rDNA and the β-tubulin dataset, indicate and support the placement of the three fungal isolates from the disinfected surface of the podetia of *Cladonia curta* within the genus *Xylaria*. The unifying morphological characters for the genus *Xylaria* include conidiophores that are usually compressed laterally into a tight layer or palisade covering all or part of the stromata surface, and conidia hyaline, ovoid to ellipsoid, with a flattened basal scar indicating the former point of attachment to the conidiogenous cell [45]. Thus, the morphological characteristics observed in the anamorphic state of these endolichenic fungi, isolated in culture, confirm their classification into this genus. The endolichenic presence of *Xylaria* representatives in *C. curta* was in agreement with previous studies that revealed a particular diversity and richness of xylariaceous fungi into lichens with diverse growth forms and substrates, in subtropical, temperate, and boreal environments [1,2,3,6,9]. 

Considering that contemporaneously endophytic and endolichenic associations are two ecologically similar interactions that influence the ecological network structure [18], both live within apparently healthy hosts and often represent the same phylogenetic lineages [1,8]. The molecular phylogeny based on the ITS region of rDNA indicated the same result as sequences of the endolichenic fungi *Xylaria* spp. recovered in this work, were found to be close to sequences of endophytic fungi cultivated from angiosperms. Likewise, these sequences were grouped together in major clades with sequences obtained from other endolichenic fungi showing topologies consistent with those reported by Arnold et al. [1] and U’Ren et al. [9], where the endolichenic and endophytic fungi of the order *Xylariales* were grouped in the same clades. This fact confirmed the endolichenic lifestyle of these species as shown by previous studies [1,2,3,6,9,32]. However, we obtained reduced bootstrap support, as well as medium support from posterior probability. 

In phylogenetic trees of the ITS region of rDNA, isolated *Xylaria* spp. were clustered with sequences of *Xylaria berteri* (Mont.) Cooke (in Index Fungorum (www.indexfungorum.org) the current name is *Xylaria berteroi* (Mont.) Cook ex J.D. Rogers & Y.M. Ju. [as ‘*berteri*’]), a species of saprophytic fungus reported as endophytic in angiosperms, but which was not registered as endolichenic previously. However, sequences from the type material of *X. berteri* are not available in the GenBank database and consequently, some authors have assigned an endophytic lifestyle to *X. berteri* based only on the BLASTn query [29,38,41]; in the GenBank database, 38 sequences of *X. berteri* were available: 27 of these correspond to sequences of 18S, 5.8S, and 28S genes and the internal transcribed spacers (ITS 1 and ITS 2) of rDNA, six sequences of the β-tubulin gene and four sequences of the genes Calmodulin, 1-alpha, alpha-actin, and DNA-dependent RNA polymerase II (as of November 2016). Notably, in the case of endophytic fungi, it is recommended that comparisons of sequence data be made using sequences from the material type of species, and if such sequences are not available, then the data must be treated with caution [46]. 

In agreement with U’Ren et al. [5], our analyses based on the β-tubulin data set revealed that the endolichenic *Xylaria* spp. could be phylogenetically related to the saprophytic *X. berteri*; however, they are more closely related to an endophytic symbiont isolated from the bark of *Cyathea lepifera* (*Cyatheaceae*) [42], suggesting that associations of endolichenic fungi with lichen thalli are not purely incidental. Although endolichenic fungi of *Xylaria* are often host-generalists, Thomas et al., [42] specified that they are not obligate and suggest foraging *Ascomycota* strategy to be a specialized survival or dispersal mechanism utilized by a subset of fungi such as *Xylaria*, and that the variation in niche or preferred habitat would modulate the selective advantage of endophytism. 

On the other hand, there are a number of indications that support the relationship between the isolated endolichenic *Xylaria* spp. from *C. curta* and *X. berteri* with a saprophytic lifestyle. First, the endolichenic fungi identified as *Xylaria cubensis* and *Xylaria* cf. *heliscus*, recovered from *L. oreinoides* and *U. mutabilis* [9] were nested within the same in-group where the endolichenic fungi of *C. curta* were located. Second, the results of a search using the BLASTn algorithm to find sequences highly similar to ours included two sequences of *Xylaria allantoidea* (Berk.) Fr. (KR534643 and KR534722) with similarity and coverage values ≥ 98%—this species is considered part of the *X. cubensis* aggregate [19,47] and was isolated from the healthy leaves of *Nertera granadensis* and *Leandra longicoma* [39]. Thirdly, *X. allantoidea*, *X.* cf. *heliscus*, and *X. cubensis*, all are located in the clade “PO” (the clade containing *X. polymorpha* and closely related species) together with *X. berteri* in the current phylogenetic status of the subfamily Xylarioideae [9,19]. Finally, was obtained a similarity in the β-tubulin alignment of our sequences from endolichenic fungi with sequences obtained from *X. berteri*, from two saprophytic fungi growing on the wood of *Castanea. carlesii* var. *sessilis* [37] and on dicot bark [19], both cultured from ascospores. In concordance, U´Ren et al. [9] suggest that in general, temperate and boreal xylariaceous endophytes, both from plants and lichens, have endophytic and saprotrophic life stages. Moreover, *Xylaria* displaying both life stages were found in the endophytic and saprotrophic phases [48].

The use of anamorphic characters of cultures often lacks sufficient information for taxonomic identification [1,32]. Furthermore, the colonial and anamorphic features described in the key by Callan and Rogers [30] for *Xylaria* species of the continental United States and Canada represent an alternative approach to the identification of lineages, but also limit the identification to the included species. Thus, the endolichenic fungi recovered in this study exhibit morphological appearance similar to the anamorphic state of five species, *X*. *multiplex*, *X*. *longiana*, *X*. *longipes*, *X*. *polymorpha*, and *X. schweinitzii*. However, is not possible to corroborate this information with other authors, since descriptions of *Xylaria* usually include descriptions of its teleomorphic state [20,31,48,49,50] and this feature was not obtained in vitro for the endolichenic *Xylaria* spp. 

The morphological features that characterize the teleomorph of *X. berteri*, including pulvinate to discoid stromata, with stipe or narrow connective and usually drooping margins (parasol-shaped) [20], differ from the features observed in the anamorphous stromata of the endolichenic fungi obtained in culture. Besides, none of the taxa of the endolichenic species isolated, present a sessile stromata, which is a defining feature of all known taxa in the *Xylaria cubensis* (Mont.) Fr. aggregate, the group that harbours penzigioid species in which *X. berteri* is probably the most frequently observed [47]. This species is reported throughout the tropics of both hemispheres to be growing on *Acacia koa*, *Albizia* sp., *Alnus nepalensis*, *Eucalyptus robusta*, *Fraxinus uhdei*, *Macadamia* sp., *Metrosideros polymorpha*, *Psidium* sp., *Sapindus saponari*, and *Spathodea campanulata* [20,51]. We also observed that the size (8)12–13.5 × 6–7.5(8) of the ascospores of *X. berteri* [20] was greater than that of the presently isolated *Xylaria* sp. 1 (2.27–2.20 × 5.01–5.10), the only species of which spores were observed. However, this observation is not conclusive considering the small number of spores found.

In vitro isolates of *Xylariaceae* specimens can be identified only after comparison with cultures that originated from identified teleomorphic materials [49]. Consequently, is not possible to determine with certainty the species-level identity of the specimen isolated from a tree and that cultured from the lichen *C. curta*, by its cultural or anamorphic features. As recommended by Stlader et al. [21] teleomorphic material is needed to establish more teleomorph-anamorph relations and finally apply a unified nomenclature to *Xylariaceae* in general using a polyphasic taxonomic approach, especially in tropical regions of the world and in the southern hemisphere. In case of a conspecific relationship between the isolated *Xylaria* spp. and *X*. *berteri*, the group of taxa reported as endophytic fungi of angiosperms and the species isolated in this work, which are recorded as endolichenic for the first time, could be a species complex. Considering that the characterization of the anamorph of *Xylaria* is required to separate closely related taxa in certain species complexes [52,53], this work provides basic information regarding characterization of the anamorph.

According to our results, the sequences of endophytic fungi from liverworts [43] that were observed to be growing inside the rhizoids without penetrating their host thalli were grouped in a separate clade of sequences of endophytic fungi of angiosperms and lichens used in the present study. Simultaneously, two endophytes in which hyphal growth was not visible in or near the rhizoids by optical microscopy are shown to integrate with the clade that includes sequences of endolichenic species from *Cladonia curta*. 

The most favoured medium for stromatal and conidial production by most *Xylaria* is 2% oatmeal agar [30], 2% MEA (Persǒh et al., 2009), or PDA (Tripathi and Joshi, 2015). However, based on the experimental observations during the progress of this work, we recommend the use of MS culture medium as an alternative for colony development of the endolichenic fungi, considering that it could favour growth. Despite this, it is necessary to use the established culture medium (i.e., MEA 2%) for the differentiation and development of structures of the anamorphic phase of these fungi in culture. 

Previous studies have already shown the variation in the expression of genes involved in symbiotic recognition in lichens, emphasizing that they have always been studied under in vitro experiment conditions for 21 stages of resynthesis (Authors). For our study we selected only those related to the first stage, being Oxidos (for fungi) and HPPD (for algae). D-arabitol dehydrogenase (Oxido) has been reported to show at least a two-fold increase in expression level in the symbiotic state [17]. The oxide gene was found over expressed in two isolates always in the first two days of re-synthesis, suggesting that this endolichenic fungi recognizes the starting point of the lichen symbiosis, even though this expression is inhibited in the days following the first encounter with the photobionte. However, the photosynthetic partner demonstrated in at least two of the tested strains an overexpression at least at some point in the resynthesis experiment. In MS1 isolate it was observed that this overexpression occurred only on the last day of cultivation and for LB1, since the first contact with the endolytic *Xylaria* species, there were overexpression peaks during the 21 days tested. Knowing that the *HPPD* gene in algae may play an important role in the synthesis of tocopherols and/or plastoquinones under stress conditions [54] our results suggest that algae did not recognize these two isolates as a possible partner for the disease or symbiosis. However, for MS2 isolate the expression of HPPD always remained low compared to Oxido, which could suggest that, under ideal conditions, this isolate and algae could form a symbiosis and lichenize, but further studies need to be done to gauge this possibility. In summary, isolates of *Xylaria* sp. isolated in the present work, apparently did not participate in the symbiosis process between *Cladonia curta* and its photobionte.

## 4. Materials and Methods

### 4.1. Collection

Samples of the lichen *Cladonia curta* were identified and collected in the municipality of São Gabriel, state of Rio Grande do Sul, Brazil, at the Federal University of Pampa (UNIPAMPA) campus (30°20’06.3” S, 54°21’46.5” W) at 124 m asl., on the dead wood from a *Eucalyptus camaldulensis* Dehnh. (Myrtaceae) trunk. The locale of collection lies within the Pampa biome located in the southern region of Brazil. The lichen sample was deposited in the Bruno Edgar Irgang herbarium (HBEI—Federal University of Pampa) under the voucher number HBEI 023.

### 4.2. Fungal Isolation and Culture 

The endolichenic fungi were isolated from the podetia of *Cladonia curta*, which was apparently healthy and without symptoms of colonization by other fungi, one day after lichen collection. Isolation of the endolichenic fungi was achieved following the spore-shot method for mycobionts [54], in which under sterile conditions, the podetia of the lichen were removed, rinsed, disinfested, and affixed to the lid of an inverted petri dish with sterile solid Vaseline (Synth^®^). Surface sterilization of the fragments was performed by consecutive immersion in 25% bleach solution for 3 min, followed by ethanol (70%), and then rinsed thrice for 2 min with ultra-pure water, and subsequently dried on sterile filter paper, following a modified version of the procedure described by Arnold et al. [1]. We used the Murashige and Skoog (MS) medium [55], modified for the growth and maintenance of endolichenic fungi. The modified MS medium (pH 5.5) contained 1.650 g L^−1^ NH_4_NO_3_; 1.9 g L^−1^ KNO_3_; 370 mg L^−1^ MgSO_4_ × H_2_O; 16.9 mg L^−1^ MnSO_4_ × H_2_O; 8.6 mg L^−1^ ZnSO_4_ × H_2_O; 0.025 mg L^−1^ CuSO_4_ × H_2_O; 333 mg L^−1^ CaCl_2_; 6.2 mg L^−1^ H_3_BO_3_; 170 mg L^−1^ KH_2_PO_4_; 0.83 mg L^−1^ KI; 0.25 mg L^−1^ NaMoO_4_ × 2H_2_O; 0.025 mg L^−1^ CoCl_2_ × 6H_2_O; 37.25 mg L^−1^ Na_2_EDTA; 27.85 mg L^−1^ FeSO_4_ × 7H_2_O; 1 mg L^−1^ thiamine; 0.5 mg L^−1^ Pyridoxine; 0.5 mg L^−1^ nicotinic acid; 2 mg L^−1^ glycine; 30 g L^−1^ sucrose; and 9 g L^−1^ agar without myo-inositol content. After 18 days, hyphae from the growth on podetia were transferred to new MS plates. The plates were maintained in a culture chamber at 20 ± 1 °C without incident light. After one month, the isolated endolichenic fungi were subcultured in MS and maintained at 20 ± 1 °C and a photoperiod of 16 h light and 8 h dark until stromata formation. To stimulate abundant stromata formation in the mycelium, the endolichenic fungi were subcultured in 2% malt extract agar (2% MEA) [55,56]. The cultures were preserved in sterile water by the Castellani method [57] and deposited at HBEI under the voucher numbers 001, 002, and 003, together with the dry stromata of isolated *Xylaria* sp. 1 and *Xylaria* sp. 2 under vouchers HBEI 021 and HBEI 022, respectively.

### 4.3. Phenotypic Characterization of Isolates in Culture Medium

We described the texture, topography, margin, exudates, coloration verse and reverse, amount of aerial mycelium, and stromata production from the colonies formed by the isolated endolichenic fungi. The stromata size, shape, surface, and colour were observed and measurements were based on the stromata available in culture. To describe the colour nomenclature, we used the dictionary of Maerz and Paul [58]. For measuring and characterizing the microscopic reproductive structures, cross-sections of the stromata were made by hand and the material was mounted in water, Melzer iodine reagent, and 5% KOH. The mean spore width and height measurement was taken from 20 randomly selected spores in water mounts. Microscopic features were measured (100× magnification) and examined by differential interface contrast (DIC) and bright-field microscopy using an optical microscope Axio Imager A2 (ZEISS^®^, Oberkochen, Germany) equipped with Axiocam ERc5s (ZEISS^®^) and software ZEN 2 v 4.0. 

### 4.4. DNA Extraction, Amplification, PCR Purification, and Sequencing

Total DNA from mycelia was extracted using a DNeasy Plant Mini Kit (Qiagen^®^, Hilden, Germany) following the manufacturer’s instructions. PCR amplification of the nuclear ribosomal internal transcribed spacers and the 5.8S region (ITS rDNA), and β-tubulin gene was performed using primers ITS1_F with ITS4_R [59] and Bt2b_F with Bt2a_R [60], respectively. The following cycle parameters were used for ITS1 and ITS4: initial denaturation at 94 °C for 2 min, 30 cycles of 45 s at 94 °C, 30 s at 55 °C, and 35 s at 72 °C, and a final elongation for 7 min at 72 °C. For the Bt2b and Bt2a the amplification conditions were: 2 min at 94 °C (denaturing), 29 cycles of 10 s at 94 °C, 10 s at 58 °C (annealing), and 20 s at 72 °C, and finally 5 min at 72 °C (extension). For the PCR reaction we used 4 μL DNA sample, 0.75 μL Milli-Q^®^ H_2_O, 1.25 μL of each primer solution, and 1.2 μL of GoTaq^®^ PCR Master Mix (Taq DNA polymerase, dNTPs, and MgCl_2_). The PCR products were purified with the Wizard PCR Preps DNA Purification System (Promega^®^, Madison, WI, USA) kit according to the manufacturer’s protocol, and quantified using the NanoVue™ Plus spectrophotometer. All products were sequenced on the ABI-Prism 3500 Genetic Analyzer (Applied Biosystems, Foster City, CA, USA). The obtained sequences were manually adjusted using the software Bioedit v. 7.2.5 [61] and a consensus sequence was obtained using the SeqMan package of Lasergene software v 14.0.0.86 (DNASTAR/Inc., Madison, WI, USA). The new sequences were deposited in the GenBank database under the accession numbers KY962975, KY962976, KY962977, MF347440, and MF347441.

### 4.5. Phylogenetic Analyses 

The ITS regions of rDNA and β-tubulin consensus sequences were compared to the GenBank database (www.ncbi.nlm.nih.gov) using the BLASTn algorithm to estimate the taxonomic placement of each isolate. The closest matched sequences with a query cover and maximum identity ≥98% and ≥97% for the ITS and β-tubulin sequences, respectively, and an e-value ≥ 0, were included in the phylogenetic analysis [62]. The available sequences (November 2016) of ITS regions of rDNA from endolichenic and endophytic fungi and the sequences for β-tubulin from endolichenic fungi reported by U’Ren et al. [9] were downloaded from NCBI. In order to provide a phylogenetic context based on ITS rDNA for the fungi isolated from *C. curta*, four clusters of sequences were established: (a) BLASTn search; (b) endophytic and endolichenic fungi reported by [1]; (c) fungi isolated from the interior of *Cladonia* and other lichen species [9] and (d) liverwort endophytic fungi cultured by David et al. [43]. For β-tubulin, one sequence group was formed including sequences resulting from the use of the BLASTn algorithm, and the sequences of endolichenic fungi reported by U’Ren et al. [9]. The GenBank accession numbers for the sequences of taxa included in the phylogenetic analyses of this study are shown in Appendix A. For each data group, the pairwise and multiple sequence alignment was performed using ClustalW implemented in MEGA v. 6.06 [63]. Models for nucleotide substitution were estimated and those with the lowest BIC scores (Bayesian Information Criterion) were considered to describe the best substitution pattern (Appendix A). Phylogenetic trees for each sequence dataset were constructed in MEGA using the Maximum Likelihood and Neighbour-Joining methods, and bootstrap values calculated from 1000 replicates (using all sites). 

From each phylogenetic tree obtained, based on the ITS regions, the sequences of endophytes and endolichenic fungi that were most closely related to the species isolated in this work were selected and the phylogenetic relationships between these were analysed using Maximum Likelihood (ML) and Bayesian Inference (BI). Maximum likelihood implemented in MEGA was used with bootstrap values calculated from 1000 replicates, the Neighbour-Joining method, and the Kimura 2-parameter substitution model. Bayesian analyses were conducted on the aligned data set using BEAST v. 1.8.3 [64], using the Tamura-Nei model of equal base frequencies, and gamma distribution with five categories. In order to identify the posterior probability tree set, 10,000,000 million Markov chains Monte Carlo (MCMC) were run and trees were sampled every 1000 generations. Tracer v1.6 [65] was used to evaluate the effective population size (ESS > 100) and TreeAnnotator v1.8.3 (from the BEAST package) was used to condense the information of trees sampled by MCMC. The phylogenies produced by ML and BI based on the ITS region of rDNA were rooted with the sequences of *Ophiostoma valdivianum* (Butin) Rulamort (NR145317), *Ophiostoma eucalyptigena* Barber & Crous (NR137979), *Ophiostoma tetropii* Math.-Käärik (NR145271), and *Peziza fascicularis* (LT158418). The phylogenetic trees for β-tubulin were constructed as described above for the ML and BI analyses. Maximum likelihood was estimated using the Tamura-Nei substitution model, and the Tamura-Nei model of equal base frequencies and gamma distribution was used for Bayesian analyses. *Ophiostoma tetropii* (AY305701), *Ophiostoma grandicarpum* (Kowalski & Butin) Rulamort (KX590762), and *Ophiostoma microsporum* Arx (KX590764) were used for rooting. The alignments and trees were deposited in TreeBase (http://purl.org/phylo/treebase/phylows/study/TB2:S21195).

### 4.6. Re-Synthesis and qPCR Evaluation on the Endophytic Role in the Lichen Symbiosis

The *Cladonia curta* photobionte was isolated in MBB medium and kept in culture for at least 30 days. After obtaining sufficient biomass the isolates were evaluated for taxonomic identification purposes and reserved for the re-synthesis tests with the fungi studied in the present work. The *Xylaria* isolates obtained in the present study were re-cultured in their original media and after reaching the entire surface of the petri dish 9 mm mycelial discs were removed and inoculated in MBB medium, then the reserved photobionte was scattered. 2.5 cm around the mycelial discs. When both isolated on the same plate, either fungal mycelium or algal cells touched, the real-time expression experiment was started.

Following the in vitro experiments, the samples were subjected to total RNA extraction protocol using the PureLink™ RNA Mini Kit (Thermo Fisher Scientific, Waltham, MA, USA). <100 mg of plant and fungi mixed tissue was used for extraction, according to the recommendations established by the kit. The extraction product was quantitated using the Qubit™ RNA HS Assay Kit (Invitrogen, CA, USA) using the Qubit Fluorometric Quantitation (Thermo Fisher Scientific, Waltham, MA, USA) kit.

A pool was performed with the replicates of each treatment, for further analysis. The samples were cDNA-transformed through the High-Capacity RNA-to-cDNATM Kit (Applied Biosystems, Foster City, CA, US) and quantitated by means of the Qubit™ dsDNA HS Assay Kit (Invitrogen, CA, USA) using Qubit Fluorometric Quantitation (Thermo Fisher Scientific, CA, USA).

Samples were diluted with Nuclease-Free Water (Promega) to standardize the amount of cDNA. The standard used was 20 ng of cDNA per mL. After standardization the qRT-PCR experiments were performed using the GoTaq^®^ qPCR Master Mix kit (Promega), the system consists of a fluorescent DNA-binding dye, the BRYT Green^®^ Dye, the protocol was used according to the instructions of the manufacturer and run on Rotor-Gene Q (Qiagen). The Oxido gene was analyzed by primer pair, Forward primer: 5’-CGTGTCAACAGTATCTCGCCTGGTA-3’, Reverse primer: 5’CACTACAGAGTAAGATGACCGCTCCTG-3’ and the HPPD gene was analyzed by primers pairs, Forward primer: 5’-GGCTCACGACATAGCAGTCAGGAA-3’, Reverse primer: 5’ACCATACAGCTCAATCTCCGACACA-3’, both previously described by Joneson et al. [17]. The expression calculated by normalizing the expression values with those of the housekeeping gene ßtub for fungi primer: Foward 5’-AGTGTGTGATCTGGAAACCTTGGAG-3’, Reverse primer: 5’-AAAGGTCATTACACAGAGGGTGCAG-3’, and chitinase-related gene for the photobionte, Foward 5’-GGCTGAGGCTTGATACACCTTGATTC-3’, Reverse primer: 5’-GCTGAAAGAGATCCTGACCAGCAACA-3’. The amplification protocol was as follows: 2 min for GoTaq^®^ Hot Start Polymerase Activation at 95 °C, followed by 45 cycles at 95 °C for 15 s, 60 °C for 1 min. For the calculation of the expression, the ΔΔCT method was used [66]. 

## 5. Conclusions

Our work highlights the less explored diversity of endolichenic fungi reported at present, including the morphological and molecular characterization of three lineages of fungi that symptomlessly inhabit the lichen thallus of *Cladonia curta*. Phylogenetic analyses based on the ITS region of rDNA and β-tubulin place these endolichenic fungi close to the genus *Xylaria*. The morphological characteristics of colonies and anamorphous stromata also confirm this classification. A phylogenetic relationship among the species of endophytic fungi identified as *X. berteri* isolated from angiosperms and the endolichenic *Xylaria* spp. is supported. However, the morphological features of the anamorphic stromata obtained in culture provide arguments that discuss identification of the endolichenic fungi isolated in this work as *X. berteri*. Given the current situation, it is premature to assign species-level taxonomic names to the isolated endolichenic *Xylaria* spp. Our preliminary data based on the molecular markers ITS rDNA and β–tubulin, provide information complementary evidence that endolichenic fungi are closely related to endophytic fungi and saprophytic fungi. Further studies, especially phylogenetic analyses using robust multi-locus datasets, are needed to accept or reject the hypothesis that the isolated endolichenic *Xylaria* spp. and *X. berteri* are conspecific. Furthermore, the use of MS culture medium constitutes an alternative in the efforts to determine the diversity of endophytic fungi that inhabit the living tissues of lichens. Finally, the diversity and prevalence of endolichenic fungi have not been studied extensively, and this is the first report of the isolation and identification of endolichenic fungi from a lichen species collected in the south of Brazil.

## Figures and Tables

**Figure 1 plants-08-00399-f001:**
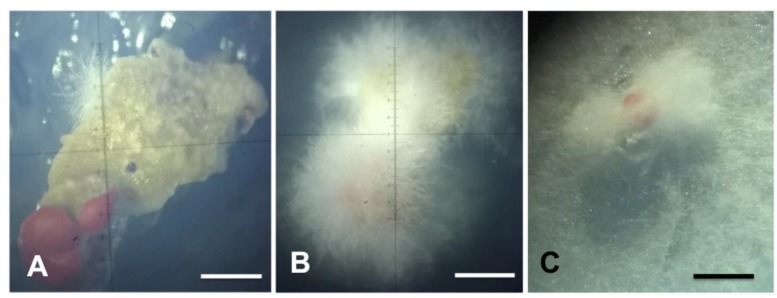
Endolichenic *Xylaria* growing on podetia of *Cladonia curta*. (**A**,**B**). Stereoscope (zoom 1.6×); (**C**). Stereoscope (zoom 3.2×); (**A**). Initial growth of hyphae on podetium after four days of inoculation; (**B**). Appearance of hyphae 18 days after isolation; (**C**). Formation of the mycelium in culture medium MS.—Scale bars (**A**–**C**) = 500 µm.

**Figure 2 plants-08-00399-f002:**
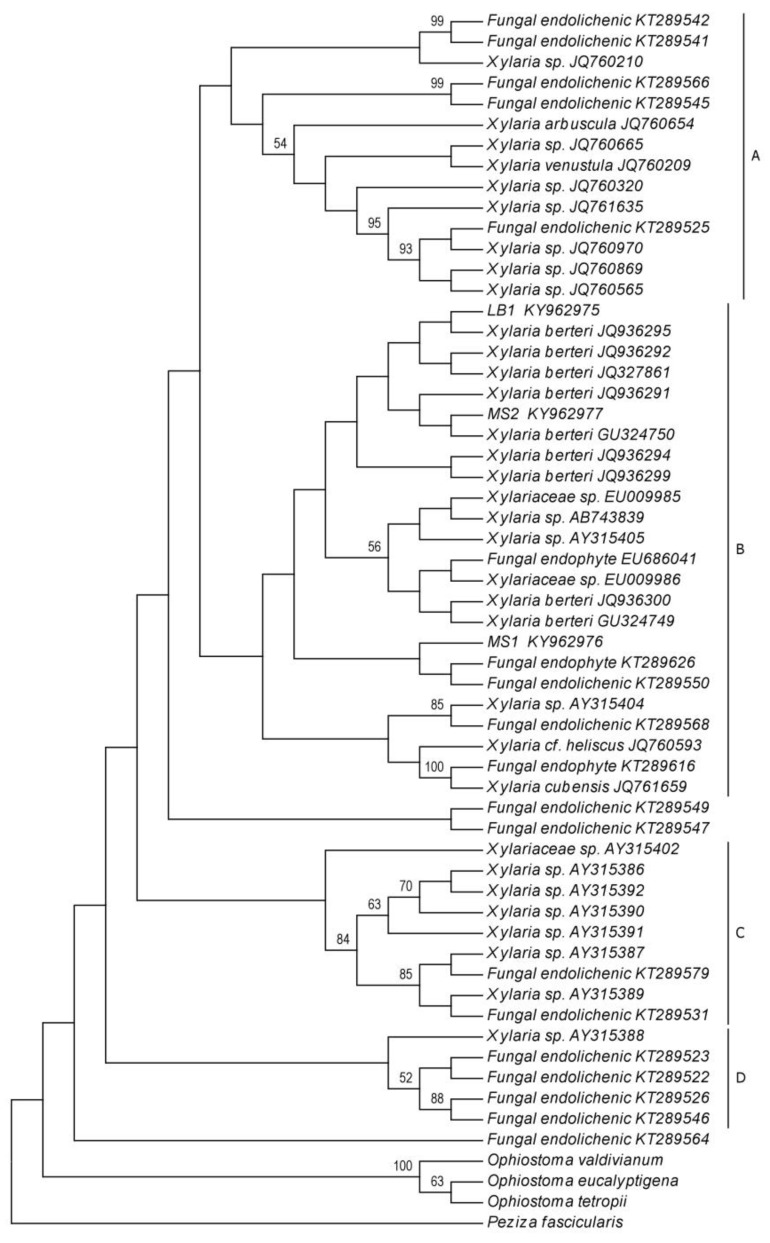
Phylogenetic relationships among endolichenic fungi isolated from *Cladonia curta* and sequences of endophytic fungi from plants (lichens [1,9], liverworts [43] and BLASTn [44]). MS1, MS2 and LB1 corresponding to sequences obtained in the present study. The phylogenetic tree inferred by maximum likelihood based on one-locus ITS region of rDNA dataset. The numbers at each internode indicates bootstrap support values. Three species of *Ophiostoma* (*O. valdivianum*, *O. eucalyptigena* and *O. tetropii*) and *Peziza fascicularis* were used as outgroup.

**Figure 3 plants-08-00399-f003:**
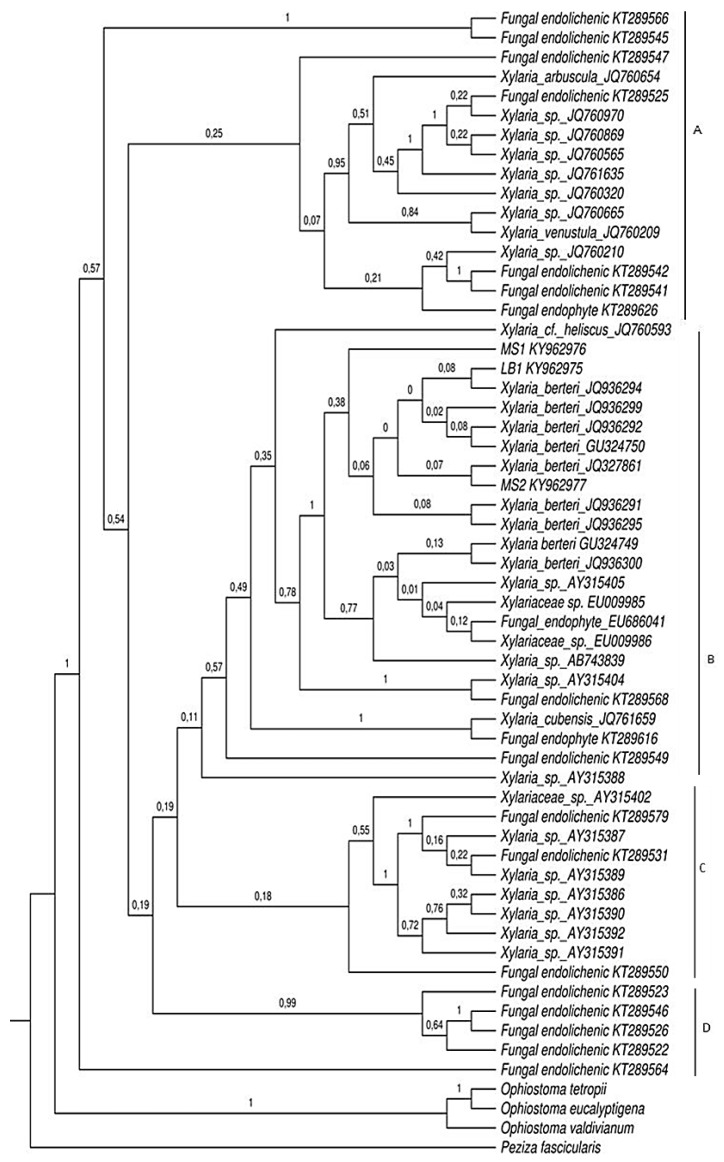
Phylogenetic relationships among endolichenic fungi isolated from *Cladonia curta* and sequences of endophytic fungi from plants (BLASTn; [44], liverworts [43] and lichens [1,9]). MS1, MS2 and LB1 corresponding to sequences obtained in the present study. A phylogenetic tree generated by using BI analysis from the Internal Transcribed Spacers ITS region of rDNA dataset. Numbers at internodes represent posterior probability values of a 95% majority rule consensus tree from 10,000,000 million generation Markov Chain Monte Carlo analysis. Three species of *Ophiostoma* (*O. valdivianum*, *O. eucalyptigena* and *O. tetropii*) and *Peziza fascicularis* were used as outgroup.

**Figure 4 plants-08-00399-f004:**
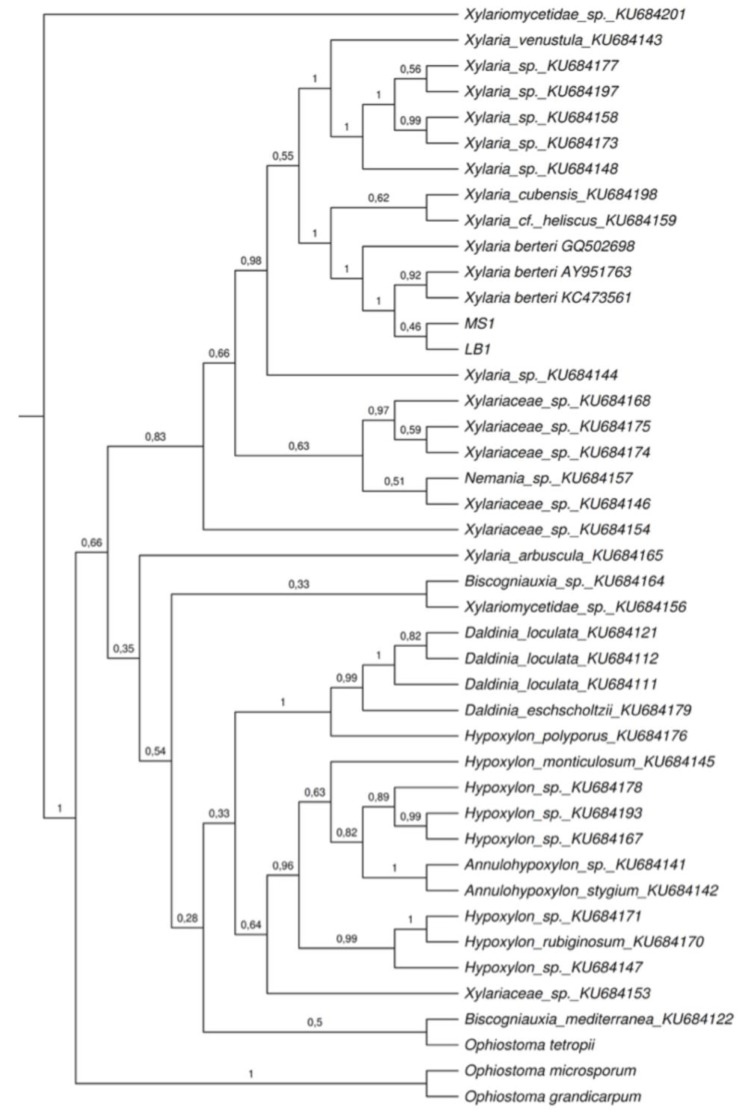
Phylogenetic relationships among endolichenic fungi isolated from *Cladonia curta*, sequences of endophytic fungi from other lichens [9] and closest match BLASTn. MS1 and LB1 corresponding to sequences of two fungi obtained in the present study. The phylogenetic tree created by using Bayesian statistical analysis with 10,000,000 million generation Markov Chain Monte Carlo based on one-locus dataset β-tubulin. The numbers at each internode indicates posterior probability values of a 95% majority rule consensus tree. *Ophiostoma tetropii*, *Ophiostoma grandicarpum* and *Ophiostoma microsporum* were used to root the trees.

**Figure 5 plants-08-00399-f005:**
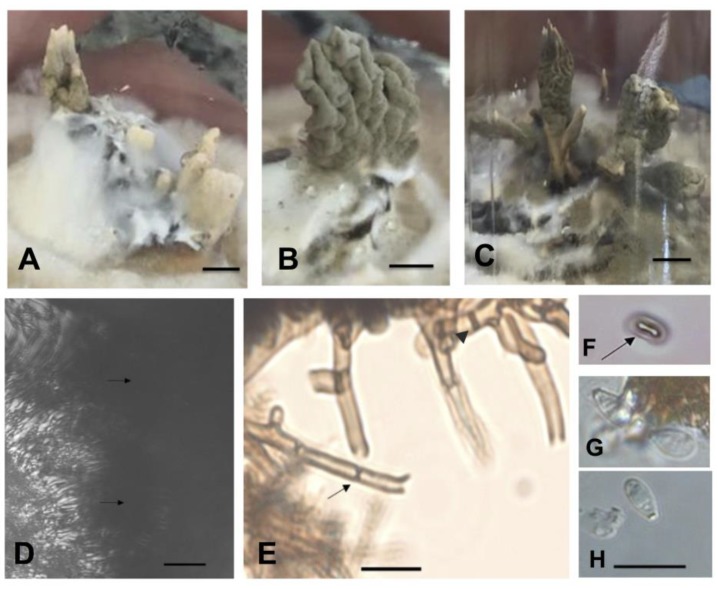
Isolated *Xylaria* sp. 1. (HBEI 021) (**A**–**C**). Stromata obtained in culture on MEA 2% medium; (**D**–**F**). Bright-field microscopy; (**D**). Initial formation perithecia (arrows); (**E**). Hyphae regularly septated (arrow) with branches forming 90° angles (arrowhead); (**F**). Ascospores with longitudinal germ slit (arrow); (**G**,**H**) Differential interference contrast microscopy (DIC), Conidiospore.—Scale bars (**A**–**C**) = 4 mm, (**D**–**H**) = 10 µm.

**Figure 6 plants-08-00399-f006:**
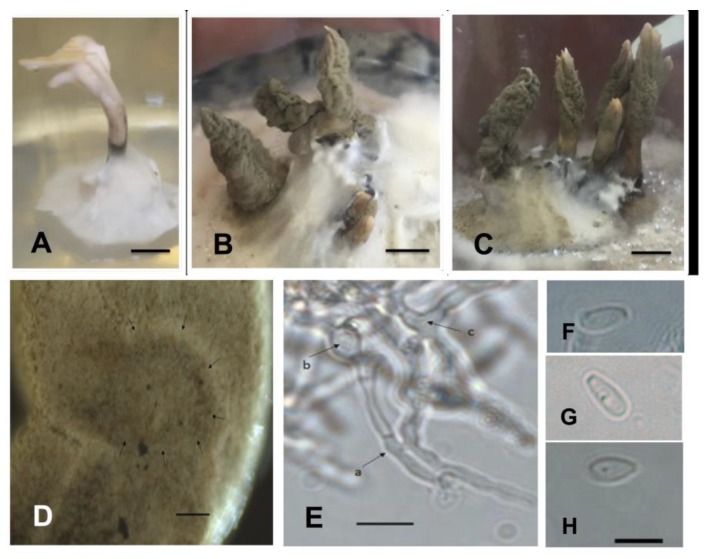
Isolated *Xylaria* sp. 2. (HBEI 022) (**A**). Stroma obtained in culture on MS medium; (**B**,**C**). Stromata on MEA 2% medium; (**D**). Initial formation of the perithecium (arrows); (**E**). Bright-field microscopy. Hyphae of hypothecium with anastomose (arrow a) and basal (arrow b) and irregular hyphae (arrow c). (**D**,**F**–**H**). Differential interference contrast microscopy (DIC); (**F**–**H**). Conidiospore.—Scale bars (**A**–**C**) = 4 mm, (**D**–**H**) = 10 µm.

**Figure 7 plants-08-00399-f007:**
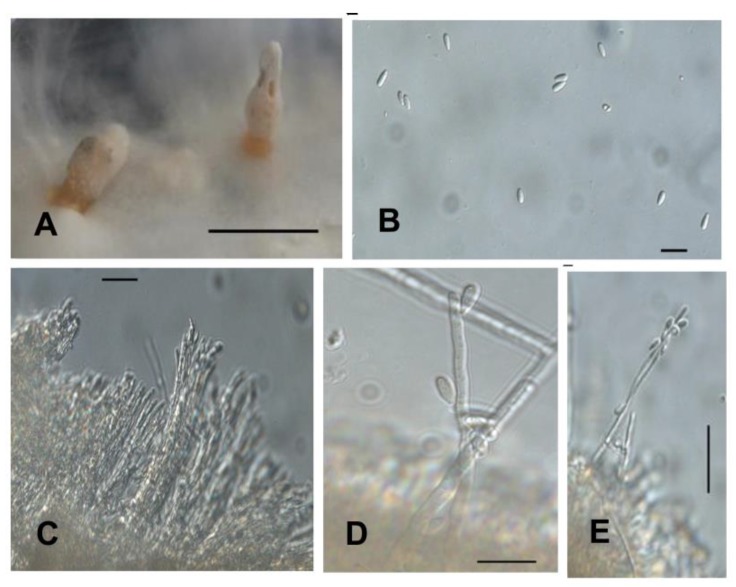
Isolated *Xylaria* sp.3. (**A**). Stromata obtained in culture on MS medium; (**B**–**E**). Differential interference contrast microscopy (DIC) (40×); (**B**). Conidiospore; (**C**). Palisade of mature conidiophores; (**D**). Mature conidiophores; (**E**). Conidia produced at the tip of a conidiophore.—Scale bars (**A**) = 1 mm, (**B**–**E**) = 10 µm.

**Figure 8 plants-08-00399-f008:**
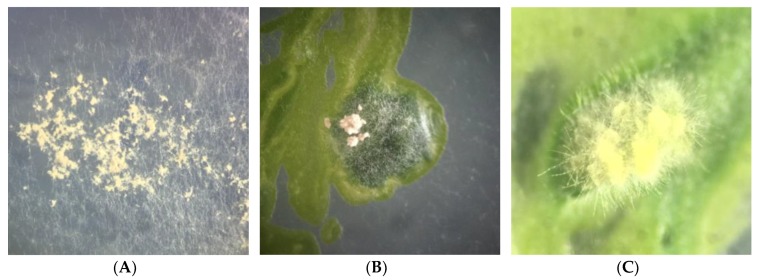
Re-synthesis evaluation in vitro: (**A**). *Xylaria* spp. mycelia growing without photobionte. (**B**). Contact of mycelium with Trebouxiaceae biomass. (**C**). Close-up showing an interaction with fungi mycelia and algae cultures.

**Figure 9 plants-08-00399-f009:**
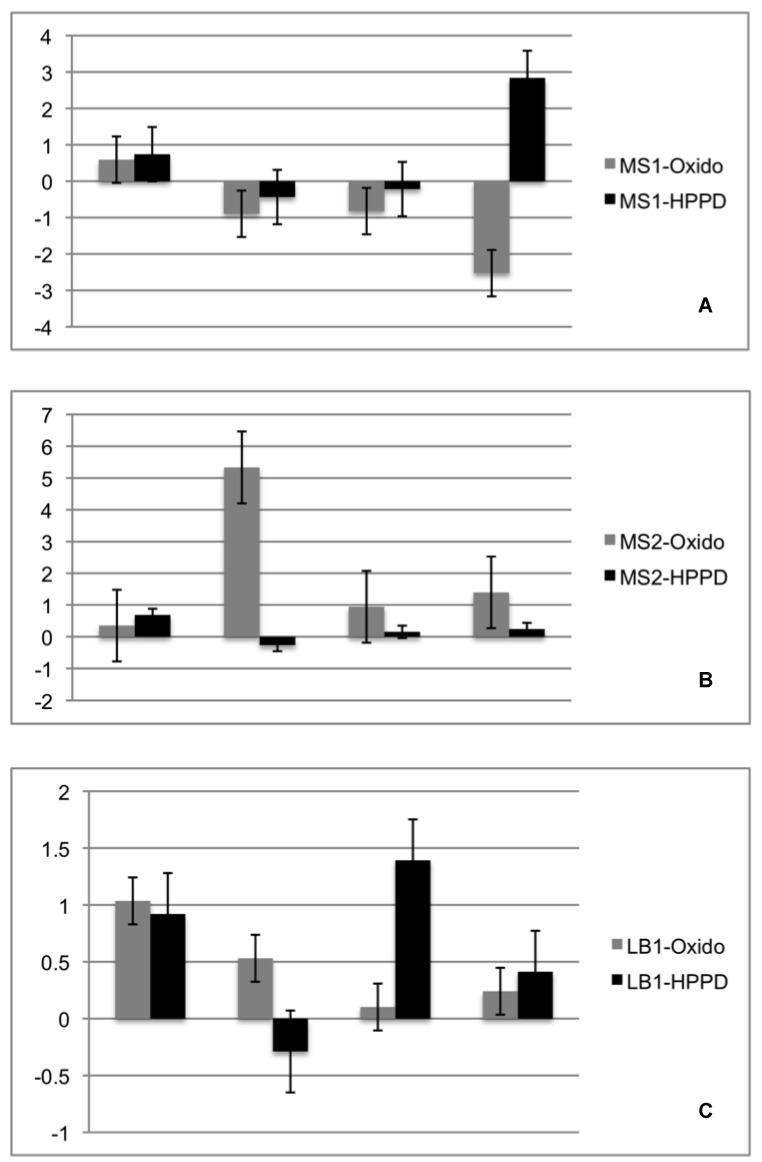
Expression Fold Change values of first stage symbiosis recognizung gene HPPD from *Xylaria* spp (black bars) and Oxido from Trebouxiaceae photobionte (grey bars) isolated from *Cladonia curta*. (**A**). *Xylaria* sp1 (MS1). (**B**). *Xylaria* sp2 (MS2). (**C**). *Xylaria* sp3 (LB1).

**Table 1 plants-08-00399-t001:** Identification of fungal endolichenic isolated from *Cladonia curta* based on ITS region of rDNA and β-Tubulin gene sequences data comparison with sequences available in the Genbank database using BLASTn algorithm. Sequences of fungi of accession numbers JF773597, KP133344, JN418792, KC771483 and HQ117853 were not included; although they met the statistical significance the studies reporting these sequences have not yet been published.

ID Isolate (Size)	Region		Sequences of Genbank					
		Nearest Match	Accession	Query Cover	Max Identity	Substrate Types	Host Species	Reference
LB1 (565 bp)	ITS rDNA	*Xylaria berteri*	GU324749	99%	98%	Bark	Dicotyledons	[19]
		*Xylaria berteri*	JQ936299	98%	99%	Leaves	*Glycine max*	[38]
		Xylariaceae sp.	EU009985	98%	98%	Leaves	*Coffea arabica*	[41]
		*Xylaria berteri*	JQ327861	99%	98%	Leaves	*Myrceugenia ovata* var. *nanophylla*	[28]
		*Xylaria berteri*	JQ936295	99%	98%	Leaves	*Glycine max*	[38]
		Xylariaceae sp.	EU009986	98%	98%	Leaves	*Coffea arabica*	[41]
		*Xylaria berteri*	JQ936294	99%	98%	Leaves	*Glycine max*	[38]
		*Xylaria berteri*	JQ936291	98%	98%	Leaves	*Glycine max*	[38]
		*Xylaria* sp.	AB743839	99%	98%	Stems	*Cinchona pubescens*	[26]
		*Xylaria berteri*	GU324750	99%	98%	Bark	Dicotyledons	[19]
MS1 (566 bp)	ITS rDNA	*Xylaria berteri*	GU324749	100%	99%	Bark	Dicotyledons	[19]
		*Xylaria berteri*	JQ327861	100%	98%	Leaves	*Myrceugenia ovata* var. *nanophylla*	[28]
		Xylariaceae sp.	EU009985	98%	99%	Leaves	*Coffea arabica*	[41]
		*Xylaria berteri*	JQ936295	100%	98%	Leaves	*Glycine max*	[38]
		Xylariaceae sp.	EU009986	98%	98%	Leaves	*Coffea arabica*	[28]
		*Xylaria berteri*	JQ936294	100%	98%	Leaves	*Glycine max*	[38]
		*Xylaria berteri*	JQ936291	99%	98%	Leaves	*Glycine max*	[38]
		*Xylaria allantoidea*	KR534643	98%	98%	Leaves	*Nertera granadensis*	[39]
		*Xylaria berteri*	GU324750	100%	98%	Bark	Dicotyledons	[19]
MS2 (515 bp)	ITS rDNA	Fungal endophyte	EU686041	99%	99%	leaves	*Plagiochila* sp.	[24]
		*Xylaria berteri*	GU324749	100%	99%	Bark	Dicotyledons	[19]
		*Xylaria berteri*	JQ936300	99%	99%	Leaves	*Glycine max*	[38]
		*Xylaria berteri*	JQ936295	100%	99%	Leaves	*Glycine max*	[38]
		*Xylaria berteri*	JQ327861	100%	99%	Leaves	*Myrceugenia ovata* var. *nanophylla*	[41]
		*Xylaria berteri*	JQ936294	100%	99%	Leaves	*Glycine max*	[38]
		*Xylaria berteri*	JQ936291	100%	99%	Leaves	*Glycine max*	[38]
		*Xylaria allantoidea*	KR534643	100%	98%	Leaves	*Nertera granadensis*	[39]
		*Xylaria berteri*	JQ936297	100%	98%	Leaves	*Glycine max*	[38]
		*Xylaria* sp.	AB743839	100%	99%	Stems	*Cinchona pubescens*	[26]
		*Xylaria berteri*	JQ936292	100%	99%	Leaves	*Glycine max*	[38]
		*Xylaria berteri*	JQ936301	98%	98%	Leaves	*Glycine max*	[38]
		*Xylaria berteri*	JQ936298	98%	98%	Leaves	*Glycine max*	[38]
		*Xylaria allantoidea*	KR534722	100%	98%	Leaves	*Leandra longicoma*	[39]
		*Xylaria* sp.	JQ341065	100%	98%	Leaves	*Diospyros crassiflora*	[34]
		*Xylaria allantoidea*	FJ884194	99%	98%	Leaves	*Hevea brasiliensis*	[35]
		*Xylaria berteri*	GU324750	98%	99%	Bark	Dicotyledons	[19]
		Fungal endophyte	EU977315	99%	98%	Twigs	Angiosperm	[40]
LB1 (404 bp)	β-tubulin	*Xylaria berteri*	AY951763	100%	98%	On wood	*Castanopsis carlesii* var. *sessilis*	[37]
		*Xylaria berteri*	KC473561	100%	98%	Bark	*Cyathea lepifera*	[42]
MS1 (409 bp)	β-tubulin	*Xylaria berteri*	AY951763	100%	97%	On wood	*Castanopsis carlesii* var. *sessilis*	[37]
		*Xylaria berteri*	KC473561	100%	97%	Bark	*Cyathea lepifera*	[42]
MS2 (242 bp)	β-tubulin	*Xylaria berteri*	FJ904911	90%	77% 1^-21^	Leaves	*Grevillea robusta*	Unpublished
		*Xylaria berteri*	AY951763	90%	77% 1^-21^	On wood	*Castanopsis carlesii* var. *sessilis*	[37]

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
