# Peer review of "Morphological and Molecular Characterization of Three Endolichenic Isolates of Xylaria (Xylariaceae), from Cladonia curta Ahti & Marcelli (Cladoniaceae)"

_plants, 2019, doi:10.3390/plants8100399_

Round 1

Reviewer 1 Report

This paper is a useful and mostly thorough contribution to the study of endophytes. It would benefit from a careful reconsideration of the resysnthesis experiment and some work on the English to ensure the meanings are clear throughout.

L75. Estimates of the genus are sometimes upwards of 1,300 species (U'ren et al. 2016)

L561. It is always curious to consider what a longer and more stringent set of methods for surface ssterilization would produce. I believe Taylor/Hyde use a longer immersion in bleach.

L126. Briefly, what were these sequences for comparison? How many isolates from how many fragments of lichen were examined? Though this is in the methods, a brief recap here would be helpful for readers.

L234. What reaction with KOH did they show?

L235. Why only 2 spores measured?

L298. Spelling of photobiont is wrong.

L297. What are you testing exactly with re-synthesis and gene expression? This is not clear in the introduction. The fungus here is not the symbiotic one from Cladonia, which is what I would expect in a resynthesis experiment for most lichens. Interesting approach, but I think that having the cladonia as part of this experiment would be informative as well; it seems possible that the expression of the xylaria with the photobiont may be misleading? Just a bit more discussion about this part of the experiment will be worthwhile. L539 is rather surprising; I wouldn't expect xylaria to lichenize with the photobiont at all. The methods don't describe this experiment adequately.

L317, etc. What does 'increasing' mean here? These photos are not very clear.

L322. How were these isolates/sequences chosen for comparison? See comment for L126.

L346. How were these isolates/sequences chosen for comparison? See comment for L126.

L380. Interaction misspelled.

L411. Figure not Figura. WHat is ithe x-axis here? Separate charts correspond to A, B, and C? These are not indicated in the figure.

L471. What genus is C. here?

Author Response

Point-by-point to reviewers

The authors met today and carefully considered the suggestions and demands of the reviewers.

In order to meet the qualification of the work, many of these suggestions were fully accepted, resulting in rewriting paragraphs for a better understanding of the questions and results obtained with the accomplishment of this work.

However, some statements by the reviewers were not considered pertinent by the authors, so that for each of them the authors offer the appropriate arguments.

A ponit-by-point presentation for all questions is presented below.:

Reviewer 1

This paper is a useful and mostly thorough contribution to the study of endophytes. It would benefit from a careful reconsideration of the resysnthesis experiment and some work on the English to ensure the meanings are clear throughout.

L75. Estimates of the genus are sometimes upwards of 1,300 species (U'ren et al. 2016)

R: This estimate presented by U'ren et al (2016) refers to the entire Xylariaceae family. The diversity estimate presented in this paper refers only to the genus Xylaria, which is the most accepted in the current literature.

L561. It is always curious to consider what a longer and more stringent set of methods for surface ssterilization would produce. I believe Taylor/Hyde use a longer immersion in bleach.

Answer: The same procedures for endolytic isolation recommended by Arnold et al (2000) and replicated by U'ren et al (2016) were used. The immersion time in bleach was chosen due to the best results and to be in line with these works cited above, thus providing reproducibility in the essays and allowing comparison with other works on the subject.

L126. Briefly, what were these sequences for comparison? How many isolates from how many fragments of lichen were examined? Though this is in the methods, a brief recap here would be helpful for readers.

Response to reviewer: The authors understand the reviewer's position, but as has been signaled by other reviewers that the paper could be shortened, we do not repeat in this paragraph what is already well placed and in detail in the methodology. Still, we rewrote for a better understanding and gave the option to visit the methodology and Table 1. which already summarizes this data.

L234. What reaction with KOH did they show?

Response to authors:  In addition to being used to rehydrate fungal samples for microscopic analysis, KOH is also used to evaluate change in pigmentation in structures. In the case of Xylaria, it is common to assess whether Perithecia shed pigments when subjected to this reagent. In our case there was no change in pigmentation and thus we only captured the image of the primordia of the perithecia when subjected to this reaction. To give the best understanding we indicated in the text that there was no change in pigmentation when subjected to 5% KOH.

L235. Why only 2 spores measured?

Response to authors: As is well known, material subjected to axenic cultivation conditions generally do not develop all structures of importance for taxonomic characterization. That way we can only get two ascopores. In the others isolated not even we can. As the authors believe that this information is valuable, obtaining ascopores (although few) under cultivation conditions, we chose to include this in the analysis and present this structure to readers. Yet we make this clearer now in that paragraph to give complete understanding.

L298. Spelling of photobiont is wrong.

Response to reviewer: That its fixed in the manuscript.

L297. What are you testing exactly with re-synthesis and gene expression? This is not clear in the introduction. The fungus here is not the symbiotic one from Cladonia, which is what I would expect in a resynthesis experiment for most lichens. Interesting approach, but I think that having the cladonia as part of this experiment would be informative as well; it seems possible that the expression of the xylaria with the photobiont may be misleading? Just a bit more discussion about this part of the experiment will be worthwhile. L539 is rather surprising; I wouldn't expect xylaria to lichenize with the photobiont at all. The methods don't describe this experiment adequately.

Response to reviewer: The authors agree with that put by the reviewer. And even other reviewers not paying attention to this, and even indicating that the paper has to be shortened, we decided to include the full explanation of the reasons for this experiment, contextualizing and giving due importance to the overall objective of this paper.

L317, etc. What does 'increasing' mean here? These photos are not very clear.

Response to reviewer: Its a typo, were fixed. Its means the zoom of Stereoscope.

L322. How were these isolates/sequences chosen for comparison? See comment for L126.

Response to reviewer: Its based in the literatura. Our sequences as homologues ITS and b-tubulin from Xylaria species, could be compared with others sequences of Xylaria endophytes well know in the literatura. For this reason we select the seqüencies published in previous papers to compare with our isolates. Its is described in the methods section.

L346. How were these isolates/sequences chosen for comparison? See comment for L126.

Response to reviewer: Based in the best hits from Blastn tool, using our sequences as a query and the database of Xylariaceae and Xylaria endophytes and endolichenic sequences available at the Genbak. Its well describe in the methods section.

L380. Interaction misspelled.

Response to reviewer: Its fixed at the manuscript.

L411. Figure not Figura. WHat is ithe x-axis here? Separate charts correspond to A, B, and C? These are not indicated in the figure.

Response to reviewer: Its fixed at the manuscript.

L471. What genus is C. here?

Response to reviewer: Its fixed at the manuscript. Its mean Castanea genus.

Reviewer 2 Report

Overall this is a well written and well researched paper. However, it has a merely descriptive aim and above all it is far too long for its scientific content. Thus, it requires to be drastically shortened before it can be accepted for publication.

In addition, but this is just an editorial choice, maybe a more lichenological (lichenologist?) of fungal journal is more appropriate for this ms.

Minor remark: I really doubt endophyte biology is limited to endolichenic fungi: the host may well be a plant and together with fungi also bacteria are found

Author Response

Reviewer 2

Comments and Suggestions for Authors

Overall this is a well written and well researched paper. However, it has a merely descriptive aim and above all it is far too long for its scientific content. Thus, it requires to be drastically shortened before it can be accepted for publication.

In addition, but this is just an editorial choice, maybe a more lichenological (lichenologist?) of fungal journal is more appropriate for this ms.

Minor remark: I really doubt endophyte biology is limited to endolichenic fungi: the host may well be a plant and together with fungi also bacteria are found.

Response to reviewer: The authors uderstand the reviewer postiton.However, to give due and broad understanding to the work, the authors did not comply with the request for shortening of the manuscript. So we would not be able to answer all the discussions resulting from our results.

Regarding the adherence of the manuscript to the journal Plants, this article has been submitted to a special edition (indicated in the submission header) that will deal especially with Lichen symbiosis. So I believe it fits in perfectly with the scope of this issue.

Reviewer 3 Report

This paper provides phylogenetic placement and describes the morphology of three strains of endlichenic Xylaria isolated from lichen-forming fungus Cladonia curta (Lecanoromycetes) collected in Brazil. However, the content of the manuscript seems to be not fully compliant with the main scope of the journal (dealing more with plants rather than fungi), it is a nice contribution to neotropical endolichenic Xylaria worth publication after considering the following points:

Recently a large phylogeny of Xylariaceae including a lot of endophytic and endolichenic Xylaria was published by U’Ren et al. (2016). In the paper, authors have used multilocus analyses (including ITS rDNA, LSU rDNA, RPB2, β-tubulin, and α-actin) to obtain sufficient supports for the tree. I would suggest adding lacking loci to the phylogenetic analyses presented in the manuscript to determined convincingly phylogenetic placement of the studied strains isolated from C. curta. The chapter 'Taxonomy' does not contain taxonomic results but instead, it provides short descriptions of strains morphology and anatomy. The specimens should be properly determined to species level or formally described if represent new species.  The message of the manuscript is in some places not clear and the language needs to be revised.

Author Response

Reviewer 3

This paper provides phylogenetic placement and describes the morphology of three strains of endlichenic Xylaria isolated from lichen-forming fungus Cladonia curta (Lecanoromycetes) collected in Brazil. However, the content of the manuscript seems to be not fully compliant with the main scope of the journal (dealing more with plants rather than fungi), it is a nice contribution to neotropical endolichenic Xylaria worth publication after considering the following points:

Recently a large phylogeny of Xylariaceae including a lot of endophytic and endolichenic Xylaria was published by U’Ren et al. (2016). In the paper, authors have used multilocus analyses (including ITS rDNA, LSU rDNA, RPB2, β-tubulin, and α-actin) to obtain sufficient supports for the tree. I would suggest adding lacking loci to the phylogenetic analyses presented in the manuscript to determined convincingly phylogenetic placement of the studied strains isolated from C. curta. The chapter 'Taxonomy' does not contain taxonomic results but instead, it provides short descriptions of strains morphology and anatomy. The specimens should be properly determined to species level or formally described if represent new species.  The message of the manuscript is in some places not clear and the language needs to be revised. 

Response to reviewer: The authors agree with that put by the reviewer. But while we are doing the experimental design and the proper review of the bibliography, we are faced with an important methodological issue. Although Data from the gene for RNA polymerase II is the largest subunit (RPB1), the gene for the minichromosome maintenance complex component 7 (MCM7) are likely to be useful to infer a complete phylonetic relationship Xylariaceae, of both endophytes and endolichenic species not currently represented in public sequence databases. Thus we had to opt for the regions where we would have enough quantitative sequences to make the proper comparisons between the isolates we found and those with the best representation of the databases.

Regarding the adherence of the manuscript to the journal Plants, this article has been submitted to a special edition (indicated in the submission header) that will deal especially with Lichen symbiosis. So I believe it fits in perfectly with the scope of this issue.

In addition, a thorough revision of English was performed, by hiring a specialized company, as certified in the attached.

Round 2

Reviewer 3 Report

The new version of the manuscript was only slightly changed and a major part of the comments proposed by all reviewers was rejected by authors. The 
authors did not answer to the point regarding chapter "taxonomy":

"The chapter 'Taxonomy' does not contain taxonomic results but instead, it provides short descriptions of strains morphology and anatomy. The specimens should be properly determined to species level or formally described if represent new species."

Author Response

The authors agree with the statements of Reviewer 3, but there are some obstacles to the real taxonomic determination of the isolates obtained in the present work.
We present below our arguments:

1. Referring to phylogenetic analyzes with the ITS region, our Xylaria are closer to endophytes identified as X. berteri whereas with the B-tubulin gene they are closer to saprophytes identified as X. berteri. In the same line of discussion, Xylaria species generally bear fruit in decayed material, but are isolated as endophytes of living hosts and that these fungi are not required, suggesting that the strategy is a specialized mechanism of survival or dispersal and that Niche variation modulates the selective advantage of endophytism. In addition, at present the “species” has the largest number of records (Taiwan, Hawaii and recently in Mexico) and the first author found an old sample recorded for Colombia. There is also talk of the species as an endophyte and they even researched the secondary metabolites produced in in vitro cultures. So could this be identified as X. berteri or at least belonging to this species complex and the manuscript would change around it entirely?

2. On the other hand we have the morphology and the characteristics of the isolates that could be simple traits of culture medium variation; however it is also well known that most species of Xylaria have typical vertical stroma except six penzigioid species: X. areolata, X. berteri, X. cranioides. crozonensis and X. frustulosa. X. berteri has sessile and atrophied stroma, and are often referred to as penzigioid species because Petch and Miller considered them to belong to the genus Penzigia, which is considered unsustainable among modern mycologists. In the report of a new species of Xylaria from Mexico (Chacón and Gonzáles, 2019), it is noted that the new species Xylaria subtropicalis of penzigioid morphology does not group with sequences from other species of penzigioides (p. Ej., X. areolata, X berteri, X. cranioids, X. discolor and X. frustulosa); they are dispersed in different branches although X. berteri and X. discolor have a morphology similar to X. subtropicalis. According to phylogenetic analyzes, the sister taxa of the new species are Xylaria longipes and Xylaria allantoidea, which differ considerably from X. subtropicalis in having stipulated fruiting bodies and spore size, indicating that there have been numerous morphological transformations within the genus.

They emphasize the same thing we talked about in the article:
“There are other factors that contribute to discrepancies between morphology and molecular data. Studies at different levels within the Xylariaceae family have identified that disagreements between morphology-based classifications and molecular phylogenies can also be attributed to the lack of molecular data from many species, including type samples, misidentified samples, and (or) database sequences. and the need for additional phylogenetic molecular markers. So our phylogeny, which is similar to many other phylogenies, reiterates the need for a complete review within the family. ” Hence, in the authors opinion would be to leave as Xylaria spp. and exchange the title of Taxonomy for "Phenotypic Characterization..." to avoid misunderstandings.

For this reason the subheadings of the item under discussion are changed to "Phenotypic characterization of isolates in culture medium"